# Depositional and Diagenetic Controls on Reservoir Quality of Neogene Surma Group from Srikail Gas Field, Bengal Basin, Bangladesh

**Maimuna Akter [1], M. Julleh Jalalur Rahman [1,*], Ming Ma [2], Delwar Hossain [1] and Farida Khanam [3]**

1 Department of Geological Sciences, Jahangirnagar University, Dhaka 1342, Bangladesh
2 North West Institute of Eco-Environment and Resources, Chinese Academy of Sciences, Lanzhou 730000, China
3 Geophysical Division, Bangladesh Petroleum Exploration and Production Company Limited (BAPEX), Dhaka 1215, Bangladesh
* Correspondence: rahman65@juniv.edu

**Abstract:** The development of an effective and profitable exploration and production depends heavily on the quality of the reservoir. The primary goal of this study was to evaluate the reservoir quality of the Neogene Surma Group at the Srikail Gas Field, which is situated in the western part of the eastern folded belt of the Bengal Basin, Bangladesh. Wire-line logs, core analysis, petrography, X-Ray diffraction (XRD) and a scanning electron microscope (SEM) were used to understand the depositional and diagenetic controls of the quality of the reservoir. The Surma Group of the Srikail Gas Field was deposited in a delta system with a dominant influence of tide. The subarkosic to sublitharenitic Neogene Surma Group sandstones have primary porosities ranging from 0% to 25.8%, with an average of 21.5%, and the secondary porosity is approximately 7%. The range of log porosity ranges from 15% to 22.2%, while log permeability and core permeability vary from 3.01 to 54.09 mD and 0.1 to 76 mD, respectively. The primary porosity had been destroyed mainly by mechanical and ductile grain compaction. Most of the clay minerals (illite/illite-smectite, chlorite and kaolinite) in sandstone occur as grain coatings, grain lining (rim) and a few occur as pore-filling. This study reveals that the reservoir quality is predominantly controlled by the depositional environment (sediment texture and facies, ductile grain supply, clay content), and diagenetic process (mainly mechanical and ductile grain compaction followed by clay cement). The information gathered from this research will be useful for future petroleum production and for enhancing predictability in order to find new prospects.

**Keywords:** petrography; diagenetic controls; reservoir quality; Neogene sandstones; Srikail gas field; Bengal Basin

## 1. Introduction

The Bengal Basin (Figure 1), one of the most outstretched sediment repositories in the world, lies in an active tectonic zone where the India, Burma and Tibetan plates intersect [1]. Bangladesh occupies a sizable portion of the Bengal Basin, and the network of rivers that traverse it deposit a huge amount of sediment. There are roughly 22 km of thick sediment from the Cretaceous to the Holocene in the Bengal Basin [2]. Among these enormous sediment areas, the Surma Group was buried to a depth of 2300 to 3100 m in more than 6000 m of Neogene sediment [2,3]. In Bangladesh, the Neogene Surma Group is thought to have been deposited in a deltaic to shallow marine environment, and has been the target of attention of geologists because the unit has reservoir sandstones alternating with sealing shales. The Surma Group is responsible for all of the oil and gas discovered in Bangladesh so far.

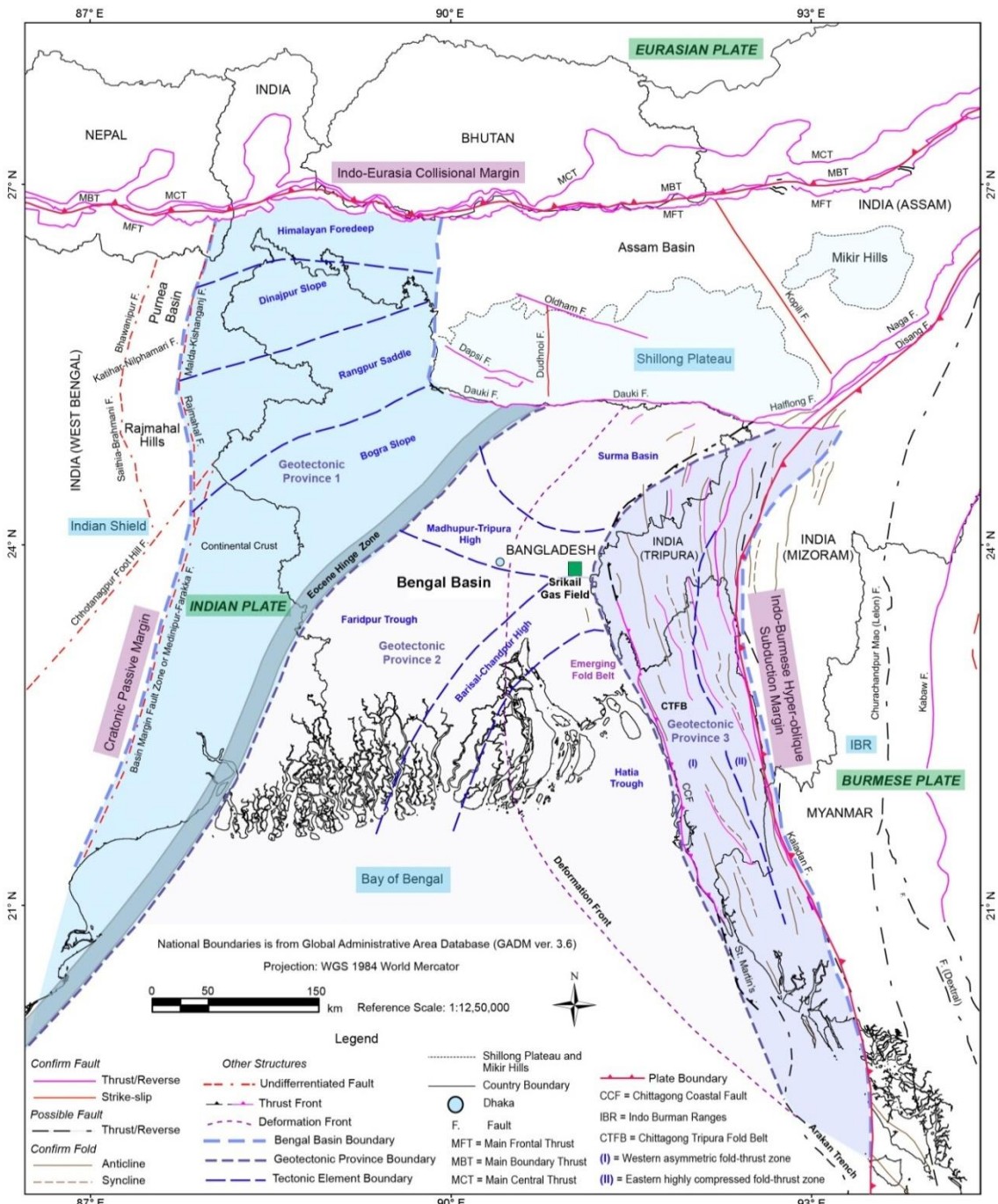

**Figure 1.** Simplified tectonic map of the Bengal Basin and its surroundings (modified after [1]) with the location of the Srikail Gas Field.

The country has proven natural gas-rich provinces in the eastern part where most of the gas fields were discovered. Among the 28 discovered gas fields, the Srikail gas field is one that lies in the western part of the fold belt of the Bengal Basin.

Recent petroleum exploration activity focusing on the fold belt part continues to add fossil fuels to the Bangladesh energy sector. Although the Srikail structure is economically significant, little attention has been paid to the reservoir quality of Neogene sandstones in this area. A crucial element in the extraction of petroleum is reservoir quality, which largely depends on the depositional environment, initial sediment composition, sand texture, and subsequent diagenetic processes [2].

However, diagenetic processes including compaction, mineral cementation, dissolution, etc. can significantly change the quality of the reservoir. In order to understand the quality of the hydrocarbon reservoir and production, it is vital to study the reservoir quality. There was only one attempt made to delineate reservoir quality of the Miocene Surma Group reservoir sandstones encountered in the Sirikail structure. The present study is based on earlier work [4] by including more petrographic data as well as reports on the diagenetic controls of the Neogene Surma Group sandstones from the Srikail Gas Field in the Bengal Basin. More new wells are being planned for the Srikail Gas Field. Therefore, it is important to have a comprehensive study on the reservoir quality of Neogene sandstones in the Srikail Gas Field.

The main purpose of this research is evaluating the reservoir sandstones using depositional and petrographic-diagenetic influences on the reservoir quality of the Neogene sandstones encountered in the Srikail Gas Field, Bengal Basin, Bangladesh.

The aim of this paper mainly is: (1) to make a detailed analysis of the depositional and diagenetic components; (2) to reconstruct the diagenetic history of the Miocene reservoir sandstones; and finally, (3) to assess the effects of depositional and diagenetic controls on the reservoir quality.

## 2. Geological Setting

The Bengal Basin (Figure 1) is a surface physiographic unit that is surrounded by the Himalayan Ranges and the Shillong Massif to the north, the Arakan-Chin Fold system of the Indo-Burman Ranges to the east, and the Indian Shield to the west. The southern border of the basin connects directly to the Bay of Bengal. The Indian and Eurasian plates collided in the north, while the Indian and Burmese plates collided in the east, creating the Bengal Basin. The formation of the Himalayan and Indo-Burmese ranges led to the establishment of the Bengal sedimentary basin [5]. During the Miocene period, the Bengal Basin was developed as a remnant ocean basin at the meeting point of the Indian plate and the Burmese plate [6].

The basin-fill at the onshore part of the basin has been divided into three geo-tectonic provinces: platform or shelf (province 1), deep-basin (Province 2) and Chittagong-Tripura Fold Belt (CTFB) (province 3) (Figure 1).The Foredeep basin is about 200 km wide to the north and gradually widens to ~500 km to the south, and shows an overall NE trend cf. [2].

The north-south trending CTFB in the eastern folded flank of the Bengal Basin was developed due to an ongoing collision between Indian Plate the Burmese Plate, exposing the Miocene to Plio-Pleistocene deposits. Tectonically the Srikail Gas Field is located on the western part of the Tripura-Chittagong fold belt [7] (Figure 1).

Major anticlines in this area generally follow a NW-SE trend with a small westward convexity. Folding becomes more severe eastward, exposing older rocks near the anticline's center. On one or both flanks, longitudinal reverse faults typically enclose each anticline. The fold belt area received a massive sediment deposit during the Cenozoic era. As this region is sandwiched between the Indian and Burmese tectonic plates, it is mostly filled with orogenic sediments that came from the eastern Himalayas and the Indo-Burma ranges to the east [1].

In terms of lithology, Neogene deposits make up the stratigraphic succession in the Srikail structure (Table 1, modified after [8]). The reservoir rocks are typically Late Miocene to Early Pliocene in age. Neogene deposits make up the majority of the deposits in the Bengal Basin. According to Evans (1932) [9], the Surma Group and the overlying Tipam Group are two significant rock units that make up the Neogene sequence in the foredeep part of the trough.

The Surma Group is assumed to have been characterized by repeated transgression and regressive cycles that were caused by tectonic subsidence as well as relative sea-level variations [10]. The Surma Group is the product of a significant delta progression since the beginning of the Neogene (Miocene) [10,11]). The Middle Miocene Bhuban Formation and the Late Miocene Boka Bil Formation are Miocene sediments that make up the Surma

Group's marine-deltaic deposits. Fluvial braided deposits are represented by the younger succession of the Pliocene braided fluvial Tipam sandstones, while fluvial meandering deposits are represented by the Late Plio-Pleistocene Dupi Tila sandstones (Table 1).

**Table 1.** Litho-stratigraphic successions of the Srikail structure (modified after [8]).

| Age | Group | Formation | Lithology | Depth (m) |
|---|---|---|---|---|
| Holocene | | Alluvium | Alluvium | Surface-50 |
| Plio-Pleistocene | Dupi Tila | Dupi Tila Sandstone | Sandstone with thin clay layers and quartzite pebbles | 50–400 |
| Pliocene | Tipam | Tipam Sandstone | Predominantly sandstone inter-bedded with thin layer of lignite, clay | 400–780 |
| Late Miocene | Surma | Upper Marine Shale | Shale with occasional intercalation of sandstone and silt | 780–900 |
| Late Miocene | | Bokabil | Alteration of sandstone and shale sequence. Predominantly shale inter-bedded with siltstone and sandstone | 900–2350 |
| Early-Middle Miocene | | Bhuban | Alternation of sandstone and thinly laminated shale sequence with minor calcareous siltstone bands. Gas saturated sand is present. | 2350–3650 (Base not seen) |

## 3. Materials and Methods

The lithological zonation, lithofacies and petrophysical characteristics have been investigated at a range of depths from 2803 to 3340 m of the Srikail-3 well (Figure 2) using cores and geophysical logs.

A total of 26 sandstone core samples were collected from the core laboratory of Bangladesh Petroleum Exploration and Production Company Limited (BAPEX) from gas-bearing zones of the D-Upper sandstone reservoir (Core-1: 3083–3092 m) and D-Lower sandstone reservoir (Core-2: 3178–3186 m) of the Srikail-3 well for petrographic study (Figure 2). Sandstone samples for thin-section study were impregnated with blue epoxy to facilitate the recognition of porosity. A research-grade polarizing microscope (Olympus BX 51, equipped an Olympus DP 72 camera) was used to study thin sections. About 600 points were counted using the point-counting method for each thin section, and modal composition, texture, as well as diagenetic features were investigated.

Clay separates (<2 μm) of shale layers within sandstones at depth 3083 m to 3090.5 m were analyzed by X-ray diffraction (XRD) using a PANalytical diffractometer with Ni-filtered Cu-K$\alpha$ radiation (40 kV, 40 mA) at the Laboratory of Soil Geology and Environment, Institute of Geology and Geophysics, Chinese Academy of Sciences (CAS). Samples for clay minerals were disaggregated in deionized water and treated with 10% $H_2O_2$ and diluted acetic acid to remove organic materials and carbonate. The particles < 2 μm were separated following Stokes' law. The clay mineral solutions were then centrifuged to concentrate clay minerals into paste and the pastes were placed on glass slides to obtain the oriented mounts. The orientated mounts were run as air-dried, heated (550 °C), and Mg-ethylene glycolated. The mineralogical composition of shale was recognized after [12–14]. The peak height above the background for each diagnostic reflection multiplied by a correction factor [15] was used for the identification of clay minerals.

The volume of shale has been computed using the linear Gamma ray (GR) method, on several zones, as usual, with a GR curve as input. The effective porosity has been computed using neutron and density logs. The permeability has been determined from effective porosity and total porosity derived from neutron and density logs. The Archie method used to calculate water saturation (Sw) is based on formation resistivity (RILD) and porosity (PHIT_ND).

The core porosity and permeability data of 14 sandstone core plugs of the Srikail-3 well and the scanning electron microscope (SEM) image was reviewed from [4].

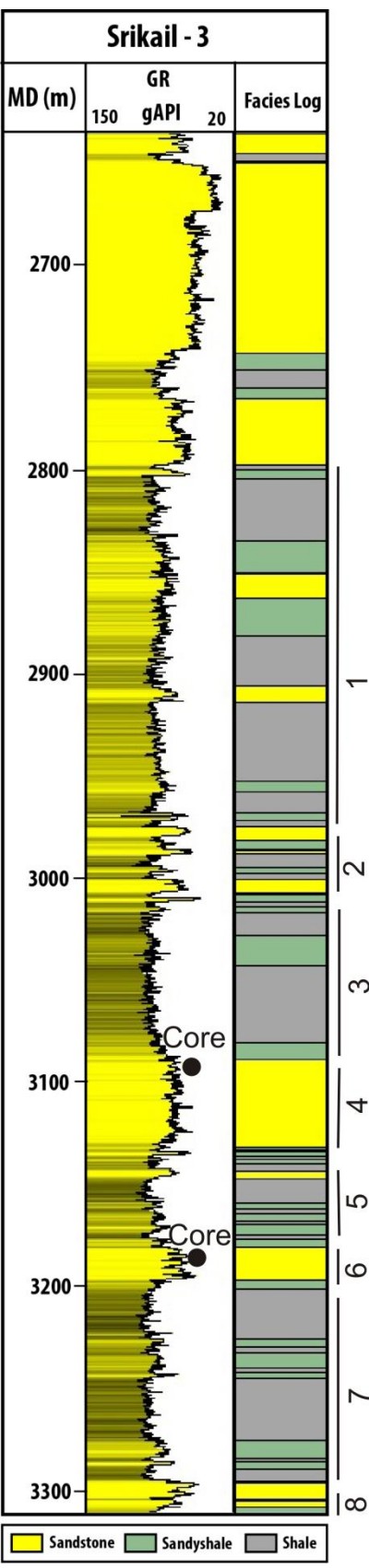

**Figure 2.** Litholog including zonation of gross lithology (1, 2, 3, 4, 5, 6, 7, 8) of reservoir and location of samples in the Srikail-3 well.

## 4. Results

### 4.1. Gross Lithology

There are a total of eight zones found in the Srikail-3 well (Figure 2). Zone 1 (2803–2968 m) is composed of mainly shale with some thin sand layers, zone 2 (2968–3016.5 m) is sandy shale, zone 3 (3016.5–3087 m) is shale, zone 4 (3087–3129.5 m) is sandstone, zone 5 (3129.5–3176.5 m) is shale, zone 6 (3176.5–3197 m) is mainly sandstone with some thin bands of shale, zone 7 (3197–3296 m) is shale, and zone 8 (3296–3321.5 m) is mainly sandstone with few shale layers.

### 4.2. Lithofacies and Facies Association

On the basis of texture, lithological association, and internal sedimentary structures, a detailed facies analysis was performed on core samples from the Srikail-3 well. Six reservoir facies, such as ripple cross-laminated sandstone (Sr), wavy-bedded sandstone (Sw), flaser-bedded sandstone (Sf), massive sandstone (Sm), planar cross-bedded sandstone (Sp), and trough cross-bedded sandstone (St) were identified from the core samples (Figure 3) which were taken from depths of 3086–3092 m and 3183–3185 m.

The principal facies of the fine grained facies association (FFA) include interbedded shale or claystone, shale with lenticular bedding, and sandstone or siltstone with shale [4].

The facies are prevalent in the Srikail-3 well from 3084 to 3085 m and 3086 to 3087 m. Claystone and shale are abundant in this facies association. Fine sandstone or siltstone lenses are observed in this facies association, and they are often connected or detached. In a quiet depositional environment with little hydrodynamic energy, this facies association forms. Because of the repeating tidal influence, they show a periodic depositional pattern [10].

Ripple cross-laminated sandstone (Sr) and wavy-bedded (Sw) and flaser-bedded (Sf) sandstone constitute the medium gained facies association (MFA). The ripple-laminated sandstone and flaser-bedded sandstone facies may have been deposited in the lower intertidal to upper subtidal zone, but the wavy-bedded facies in this facies association is thought to have been deposited under moderate to low energy conditions in the middle portion of the intertidal zone [16]. Sedimentary structures seen in this facies association suggest that they may have been formed in the intertidal zone, which is developed by the progradation of tidal flats. Massive sandstone (Sm), planar cross-bedded sandstone (Sp) and trough cross-bedded sandstone (St) make up the coarse grained facies association (CFA). The medium to coarse grain size of this facies association suggests that there is a strong traction current transporting the sediments. Massive structure and mud clasts indicate channel fill deposits at high energy. In the upper flow regime of a shallow tidal channel, parallel cross-bedded sandstones are deposited under high energy conditions. A high velocity turbulent flow is also responsible for these deposits [10]. Cross-bedded sandstones may be formed by a large migrating bedform that is controlled by high tidal energy in a broad subtidal environment.

According to a detailed investigation of several facies, characteristic sedimentary structures and facies associations, it can be revealed that the sediments of the Surma Group in the Srikail structure were deposited in a subtidal and intertidal environment of a tide-dominated deltaic setting.

### 4.3. Petrographic Characteristics of Sandstone

The petrographic results are summarized in Table 2. An analysis of 28 thin sections of sandstones reveals that the grains are mostly quartz, varying from 32.5% to 51.1%, with an average of 41.6%. Among them, monocrystalline quartz (Figure 4) is averages 34.9% and polycrystalline quartz grain (Figure 4) averages 6.8%. Feldspar (Figure 4a,d) makes up an average of. 3.1%, of which plagioclase feldspar is about 1.1% and K-feldspar is approximately 2.1%. Lithic grains, which account for an average of 9% of the total frameworks in the samples analyzed, are the most prevalent detrital rock particles after quartz. Chert (Figure 4c), shale (Figure 4f) and siltstone are prominent sedimentary lithic grains that make up an average of about 6.5%. The low-grade metamorphic lithic grains, such

as graphite-mica-schist, mica-schist, and quartz-mica-schist that are commonly observed (Figure 4a,c) that constitute an average of 2.5% of the sandstone. Biotite (av. 3.3%) was found to be more predominant than white mica (av. 2.7%). Biotite is found frequently altered to chlorite (Figure 4c). Of the total framework grains, carbonate makes up approximately 0.18% and chlorite constitutes about 0.46%. The average content of the matrix in sandstone samples is around 1.6%.

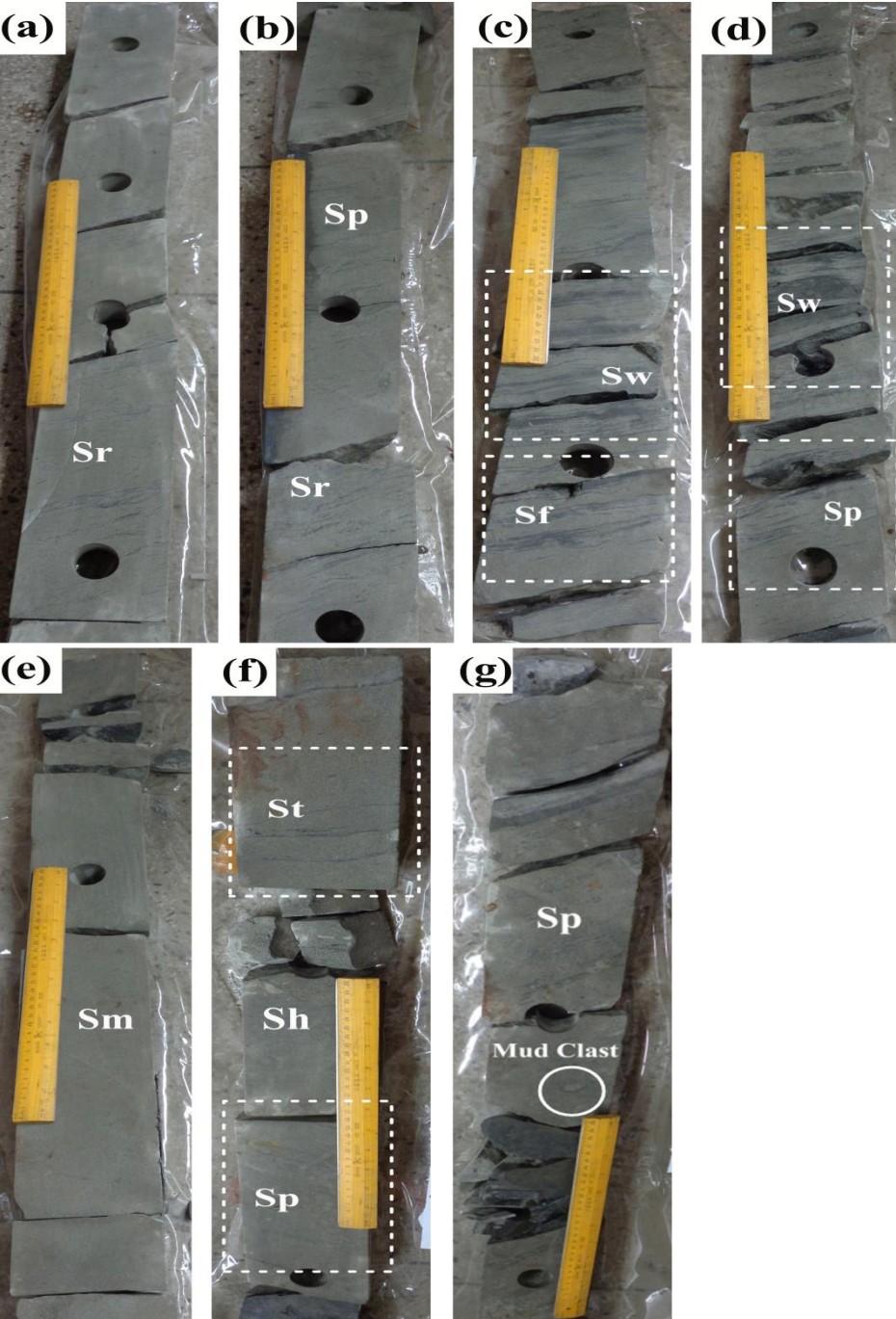

**Figure 3.** (**a**) Ripple cross-laminated sandstone (Sr) (3086–3087 m); (**b**) Planar cross-bedded sandstone (Sp) and ripple cross-laminated sandstone (Sr) (3087–3088 m); (**c**) Wavy-bedded sandstone (Sw) and flaser-bedded sandstone (Sf) (3089–3090 m); (**d**) Wavy- bedded sandstone (Sw) and planar cross-bedded sandstones (Sp) (3090–3091 m); (**e**) Massive sandstone (Sm) (3091–3092 m); (**f**) Trough cross-bedded sandstone (St), Parallel bedded sandstone (Sh) and Planar cross-bedded sandstone (Sp) (3183–3184 m); (**g**) Planar cross-bedded sandstone (Sp) with mudclast (3184–3185 m).

Table 2. Framework composition of the Neogene reservoir sandstones observed in the Srikail-3 well (expressed as percentage).

| Sample No. | Depth (m) | Qm | Qp | Chert | K | P | Biotite | W. Mica | D. Chl | D. Car | Matrix | Ls | Lv | Lm | Chc | Clc | Cc | Qc | Pp | Ps |
|---|---|---|---|---|---|---|---|---|---|---|---|---|---|---|---|---|---|---|---|---|
| 1a | 3083.5 | 36.5 | 7.4 | 7.09 | 2.25 | 1.1 | 2.8 | 1.57 | 0.47 | 0.31 | 1.26 | 0.16 | 0 | 3.15 | 0.47 | 6.93 | 0.94 | 0.31 | 20.6 | 6.3 |
| 1b | 3084 | 37.1 | 6.3 | 5.79 | 1.66 | 0.33 | 3.8 | 1.66 | 0 | 0.83 | 3.48 | 0.17 | 0 | 0.83 | 4.97 | 8.11 | 0.33 | 0 | 18.9 | 5.79 |
| 2a | 3084.5 | 42.3 | 8.8 | 6.45 | 0.99 | 0.33 | 3.5 | 2.64 | 0.17 | 1.32 | 1.49 | 0.17 | 0 | 0.5 | 1.32 | 6.28 | 0.99 | 0 | 17.5 | 5.29 |
| 3a | 3085 | 38 | 4.7 | 6 | 3.17 | 1 | 3.8 | 3.83 | 0.17 | 0.83 | 1.33 | 0.67 | 0 | 4.83 | 0.67 | 3 | 0 | 0.33 | 19.7 | 8 |
| 3b | 3085.5 | 27.5 | 6 | 3.67 | 3.5 | 1.33 | 6.7 | 4.5 | 0.17 | 1.33 | 2 | 0.5 | 0 | 5.5 | 1.33 | 4.17 | 8.5 | 0.5 | 16.2 | 7 |
| 3c | 3086 | 34.5 | 5.2 | 4 | 3.33 | 1 | 3.2 | 2.5 | 0.67 | 0.33 | 1.5 | 0.17 | 0 | 5.83 | 2.17 | 6 | 1.17 | 0.83 | 20.5 | 7.83 |
| 4 | 3086.5 | 29.6 | 4.6 | 5.43 | 1.48 | 1.64 | 11 | 5.26 | 0.16 | 0.33 | 1.81 | 0.16 | 0 | 3.45 | 0.16 | 10.9 | 0.49 | 0.16 | 17.4 | 6.09 |
| 5a | 3087 | 27.4 | 5.1 | 5.8 | 2.99 | 1.99 | 3.5 | 3.65 | 0.17 | 1 | 0.83 | 0.66 | 0 | 4.15 | 0.5 | 8.29 | 1.33 | 0.17 | 23.7 | 8.79 |
| 5b | 3088 | 32.8 | 5.1 | 7.13 | 3.32 | 1.49 | 2 | 2.82 | 0.17 | 1 | 0.66 | 0.5 | 0 | 2.49 | 1.66 | 4.64 | 0 | 0.33 | 25.2 | 8.62 |
| 6 | 3089 | 31.8 | 6.8 | 5.46 | 1.49 | 0.66 | 5.6 | 5.13 | 0 | 0.66 | 1.66 | 0.17 | 0 | 2.98 | 0.66 | 6.46 | 0.66 | 0.5 | 21.9 | 7.45 |
| 7a | 3089.5 | 33.7 | 5.5 | 6.12 | 2.42 | 1.61 | 6 | 2.58 | 0.48 | 0.32 | 1.61 | 0.81 | 0 | 3.06 | 0.97 | 8.7 | 0.32 | 0.32 | 20.3 | 5.31 |
| 7b | 3090 | 28.2 | 6.5 | 6.54 | 2.07 | 1.28 | 6.2 | 4.63 | 0.32 | 1.12 | 1.59 | 0.48 | 0 | 3.19 | 0.8 | 10 | 1.44 | 0 | 20.3 | 5.26 |
| 8a | 3090.5 | 29 | 6.9 | 6.43 | 1.81 | 1.48 | 2.6 | 2.8 | 0 | 0.82 | 1.15 | 0.66 | 0 | 2.64 | 0.66 | 9.56 | 0.49 | 0.33 | 24.9 | 7.74 |
| 9a | 3091 | 37.2 | 7.1 | 7.27 | 2.48 | 0.99 | 1.5 | 1.49 | 0.33 | 0.33 | 0.99 | 0 | 0 | 2.31 | 1.16 | 4.13 | 0.5 | 0 | 25.3 | 6.94 |
| 9b | 3092 | 34.7 | 4.2 | 7.14 | 2.49 | 1.16 | 3.3 | 4.49 | 0 | 1.5 | 1 | 0.33 | 0 | 2.82 | 0.66 | 3.99 | 1.16 | 0.5 | 22.8 | 7.81 |
| 10a | 3177.5 | 33.6 | 9.3 | 6.64 | 2.16 | 1 | 2.2 | 1.99 | 0.17 | 0 | 1.5 | 0.66 | 0 | 1.99 | 2.66 | 3.32 | 0.17 | 0 | 24.6 | 8.14 |
| 10b | 3178 | 35.4 | 7.2 | 6.59 | 2.25 | 0.96 | 1 | 1.93 | 0.48 | 0 | 2.57 | 0.32 | 0 | 1.61 | 3.38 | 3.38 | 0 | 0.32 | 25.6 | 7.07 |
| 10c | 3179 | 37 | 7.6 | 6.78 | 1.49 | 1.32 | 1 | 1.82 | 0 | 0 | 1.16 | 0.17 | 0 | 3.31 | 3.64 | 2.15 | 0.17 | 0.17 | 24.8 | 7.44 |
| 11a | 3179.5 | 38.1 | 3.3 | 3.59 | 0.16 | 0.82 | 0.7 | 1.31 | 0 | 0 | 0.49 | 0 | 0 | 0.49 | 0.65 | 4.08 | 46.4 | 0 | 0 | 0 |
| 11b | 3180 | 38.2 | 7.7 | 6.33 | 2.33 | 0.83 | 1 | 1.33 | 0.33 | 0 | 2.83 | 0 | 0 | 1.5 | 1.83 | 2.33 | 0 | 0 | 25.5 | 8 |
| 12 | 3181 | 40.3 | 5 | 4.83 | 1.33 | 0.67 | 4.8 | 3.33 | 0 | 0 | 1.83 | 0.17 | 0 | 1.17 | 2.5 | 2.83 | 0 | 0 | 24 | 7.17 |
| 13 | 3181.5 | 36.9 | 8.1 | 5.63 | 1.82 | 0.66 | 1.8 | 1.32 | 0.17 | 0 | 1.32 | 0.17 | 0 | 1.49 | 2.81 | 4.14 | 0 | 0 | 25.5 | 8.11 |
| 14 | 3182 | 37.9 | 8.4 | 7.06 | 1.48 | 0.66 | 3.8 | 1.15 | 0.16 | 0 | 1.64 | 0.16 | 0 | 1.15 | 3.28 | 1.31 | 0.49 | 0.16 | 25 | 6.24 |
| 15 | 3183.5 | 36.3 | 11 | 6.67 | 1.5 | 1 | 0.8 | 1.67 | 0 | 0 | 2.17 | 0.17 | 0 | 1.83 | 3.33 | 2.33 | 0 | 0 | 25.8 | 5.17 |
| 16a | 3184.5 | 36.7 | 9.2 | 7.58 | 1.65 | 0.82 | 1.6 | 2.14 | 0 | 0 | 1.65 | 0.16 | 0 | 2.31 | 2.64 | 2.64 | 0 | 0 | 24.2 | 6.59 |
| 17 | 3185.5 | 35.8 | 9.1 | 8.1 | 1.62 | 1.46 | 2.6 | 2.27 | 0.16 | 0 | 2.76 | 0.65 | 0 | 1.46 | 3.24 | 2.59 | 0 | 0 | 20.3 | 7.94 |
| Average | | 34.9 | 6.8 | 6.16 | 2.06 | 1.06 | 3.3 | 2.69 | 0.18 | 0.46 | 1.63 | 0.32 | 0 | 2.54 | 1.85 | 5.09 | 2.52 | 0.19 | 21.5 | 6.77 |

Note: Qm—monocrystalline quartz, Qp—polycrystalline quartz, K—potassium feldspar, P—plagioclase feldspar, Ls—sedimentary rock fragments, Lv—volcanic rock fragments, Lm—metamorphic rock fragments, Chc—chlorite cement, Clc—clay cement, Cc—calcite cement, Qc—quartz cement, Pp—primary porosity, Sp—secondary porosity, D. Chl—detrital chlorite, D. Car—detrital carbonate.

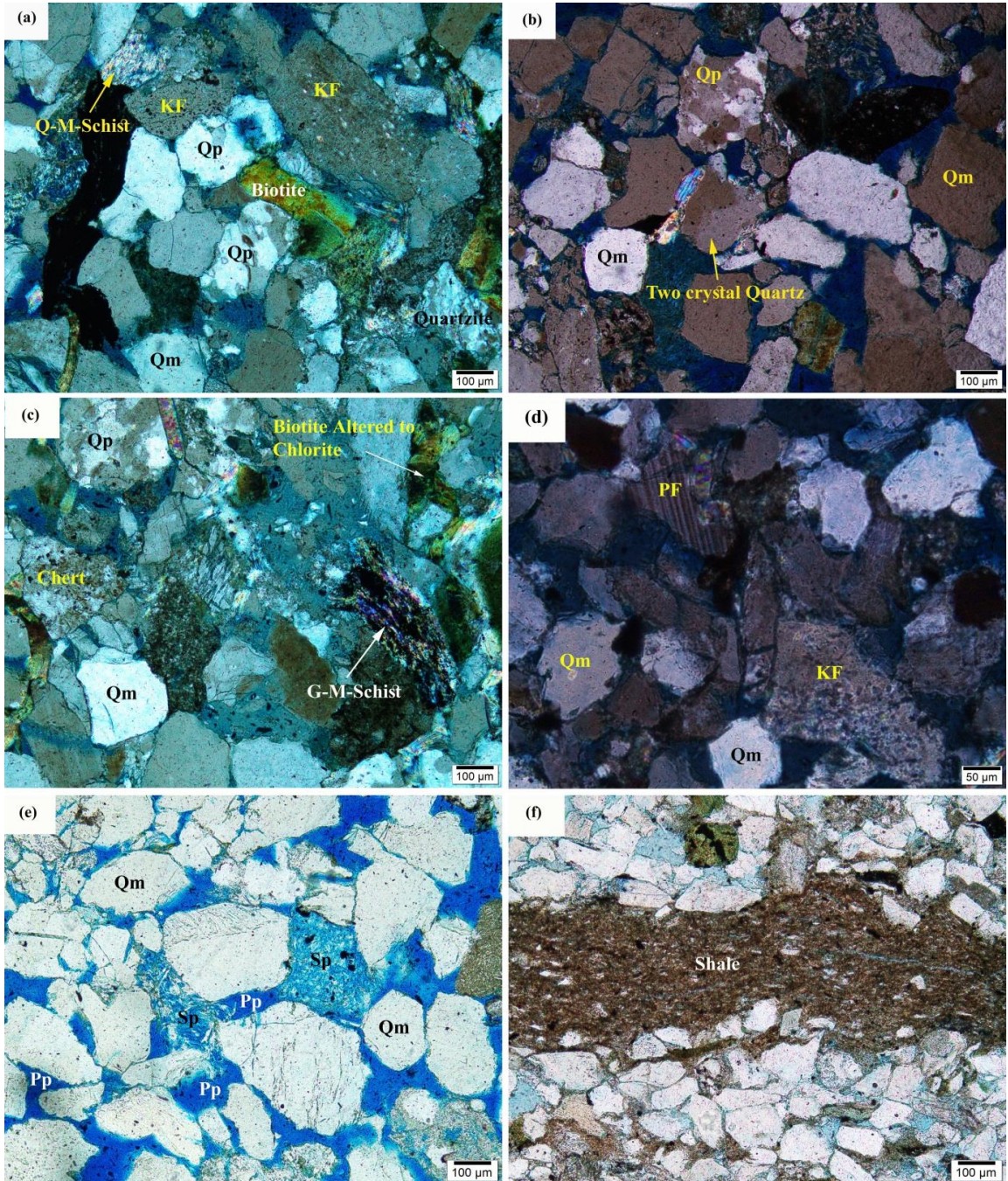

**Figure 4.** Photomicrographs of the Srikail-3 well: (**a**) showing metamorphic lithic grain (Quartz-mica-Schist, Q-M-Schist), monocrystalline quartz, Qm, polycrystalline quartz, Qp, Potassium feldspar, KF, depth 3182 m; (**b**) monocrystalline quartz, Qm, polycrystalline quartz, Qp, blue areas (pore spaces) depth 3180 m; (**c**) showing metamorphic lithic grain (Quartz-mica-Schist, Q-M-Schist), monocrystalline quartz, Qm, polycrystalline quartz, Qp, chert, biotite being altered to chlorite-depth 3182; (**d**) showing monocrystalline quartz, Qm, Potassium feldspar, KF, Plagioclase feldspar, PF-depth 3089 m; (**e**) showing monocrystalline quartz, Qm, Primary porosity (Pp), Secondary porosity (Sp), depth 3183.5 m; (**f**) shows shale interlayering within sandstone—depth 3088 m.

According to Folk's (1974) [17] categorization, the sandstones of the Srikail-3 well are mainly of sublitharenite to feldspathic litharenite (Figure 5a) and by McBride's classification (1963) [18], the sandstones are of sublitharenite to subarkosic, while a few are quartz arenite (Figure 5b) in nature.

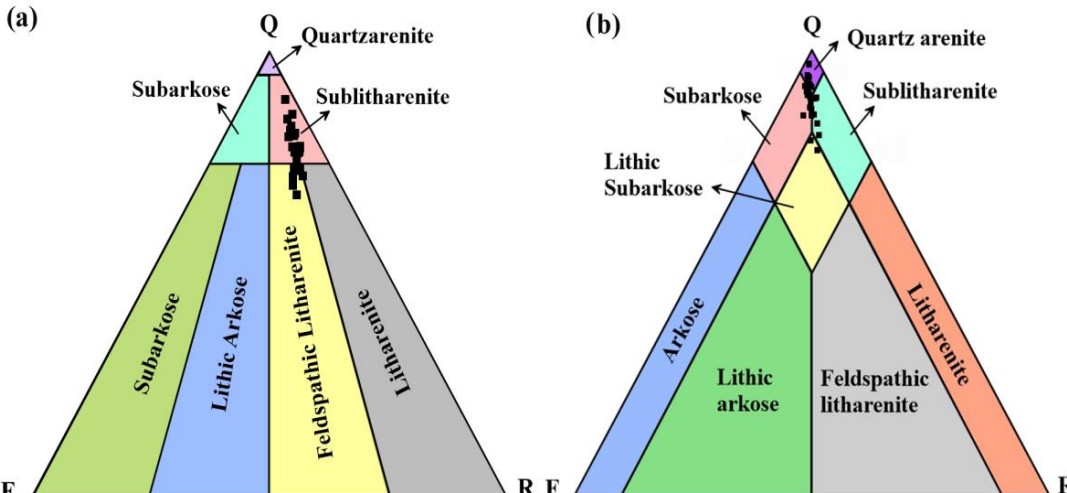

**Figure 5.** QFR (Quartz-Feldspar-Rock fragments) triangular plot for classification of the studied sandstone samples of Srikail well-3 (**a**) after Folk (1974) [17] and (**b**) after McBride (1963) [18].

### 4.4. Petrophysical Properties

The petrophysical parameters estimated from well logs are presented in Table 3 and Figure 6. The total porosity value ranges from 6 to 35%, where the average effective porosity (PHIE) through all reservoir units is 17.76%. The maximum log effective porosity is 22.2%, which is noted to be between 3129.5 and 3176.5 m in depth, and the lowest log effective porosity is 15%, which is found between 3087 and 3129.5 m depth (Table 3). The average value for log permeability is 20 mD. In the depth range of 3129.5–3176.5 m, a higher value of log permeability up to 54 mD was observed (Table 3). Shale volumes (VSH) ranged from 9% to 15%, with an average of 11.4%, showing the comparatively clean nature of sandstones. The range of the water saturation (SW) was from 0% to 49%, indicating that the gas sands are thoroughly saturated with hydrocarbons.

**Table 3.** Petrophysical properties using well log data from Srikail well-3.

| Depth (m) | Av. Shale Volume (VSH) | Av. Water Saturation (SW) | Av. Porosity (%) | Av. Permeability (mD) |
|---|---|---|---|---|
| 2830–2968 | 0.11 | 0.0 | 16.8 | 7.35 |
| 2968–3016.5 | 0.13 | 0.0 | 18.8 | 8.73 |
| 3016.5–3087 | 0.10 | 0.34 | 16.3 | 5.23 |
| 3087–3129.5 | 0.12 | 0.39 | 15 | 3.65 |
| 3129.5–3176.5 | 0.09 | 0.48 | 22.2 | 54.09 |
| 3176.5–3197 | 0.09 | 0.29 | 15.2 | 3.01 |
| 3197–3296 | 0.11 | 0.49 | 18.8 | 16.76 |
| 3296–3321.5 | 0.15 | 0.44 | 15.9 | 27.7 |
| 3321.5–3340 | 0.13 | 0.35 | 20.8 | 52.83 |

The point-counted thin section primary porosity (intergranular) ranges from 0% to 26%, with an average of 21%, and secondary porosity (intragranular, over-sized and mold) varied from 0 to 8.8%, with an average of 6.7% (Table 2). Here, the maximum porosity value was found at a depth of 3088 m and a minimum value at 3179.5 m depth. The average core porosity is 15.68%, with a range of 7.44% to 18.56% [4]. With an average permeability of 31.07 mD, core permeability ranges from 0.1 to 76 mD (Table 4). Core porosity and permeability both had the lowest values at 3084 m depth, but maximum values were found at depths of 3181 m and 3182 m, respectively [4]. The wide variation in permeability may be due to mechanical and ductile grain compaction and cementation. Overall, petrophysical characteristics revealed that reservoir sandstones of the Srikail-3 well have fair to good

porosity (>15%), and poor to moderate permeability values of permeability (0.1–76 mD, av. 31 mD) offered moderate to good quality in the reservoir.

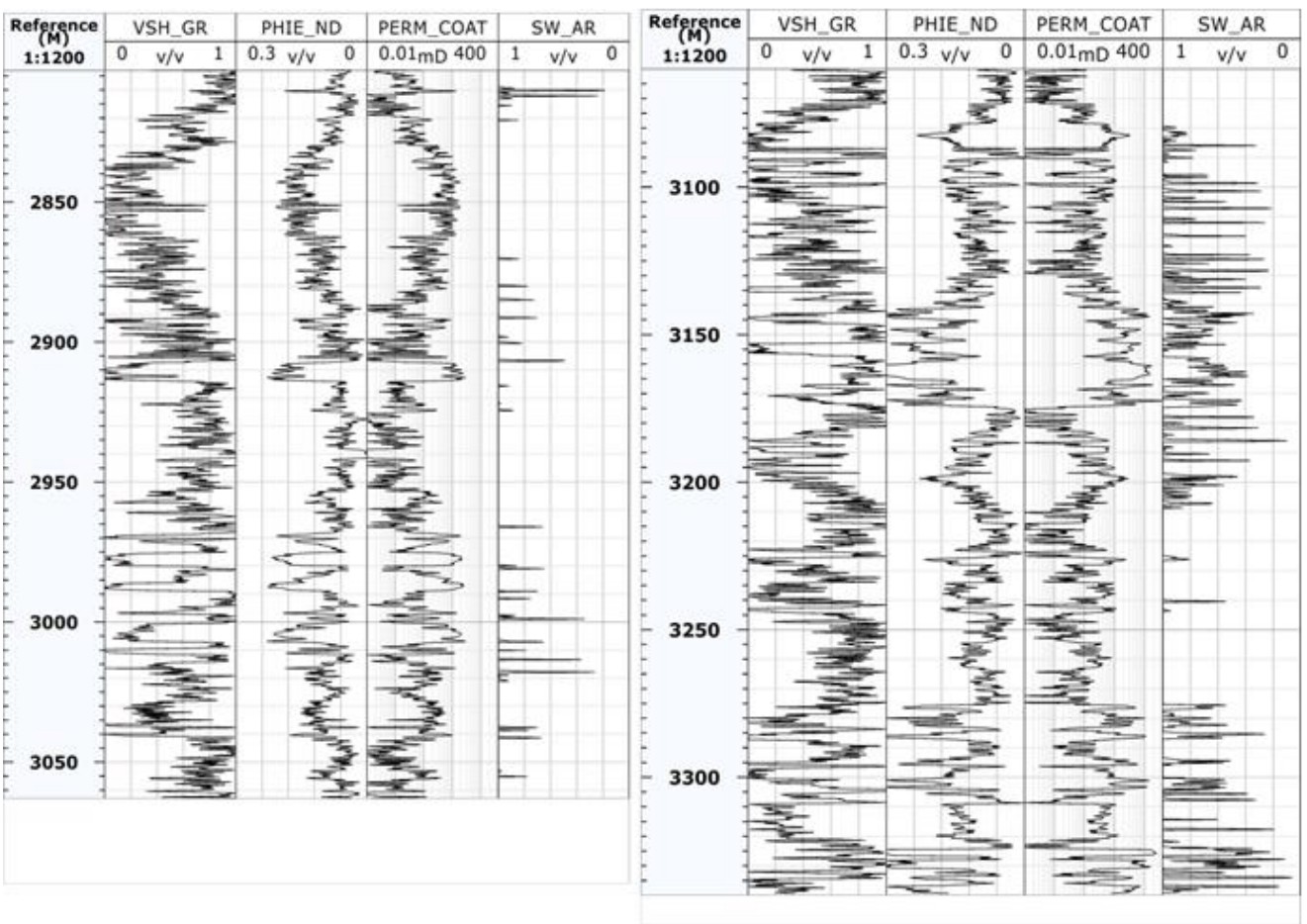

**Figure 6.** Well log showing shale volume, porosity (Neutron- density log), permeability and water saturation values of various depths from the Srikail-3 well. VSH_GR-shale volume from Gamma ray log; PHIE_ND-Effective porosity from Neutron Density log; PERM_COAT-Permeability after Coate's method; SW_AR-water saturation following Archie equation.

**Table 4.** Core Porosity and permeability data from Srikail well-3.

| | Core porosity and Permeability [4] | | |
| Sample | Depth (m) | Porosity (%) | Permeability (mD) |
|---|---|---|---|
| S-3-3084 | 3084 | 7.44 | 0.1 |
| S-3-3085 | 3085 | 10.58 | 0.67 |
| S-3-3087 | 3087 | 18.56 | 73.24 |
| S-3-3088 | 3088 | 16.15 | 4.24 |
| S-3-3089 | 3089 | 17.85 | 33.03 |
| S-3-3090 | 3090 | 17.21 | 4.77 |
| S-3-3091 | 3091 | 15.55 | 2.46 |
| S-3-3179 | 3179 | 15.72 | 33.31 |
| S-3-3180 | 3180 | 15.71 | 10.03 |
| S-3-3181 | 3181 | 18.17 | 22.61 |
| S-3-3182 | 3182 | 17.44 | 75.69 |
| S-3-3183 | 3183 | 16.98 | 71.24 |
| S-3-3185 | 3185 | 17.02 | 68.18 |
| S-3-3186 | 3186 | 15.2 | 35.45 |

The porosity and permeability of the plugged core, as well as the wireline log, show that there is a positive relationship between porosity and permeability (Figure 7a,b). Permeability rises in conjunction with porosity.

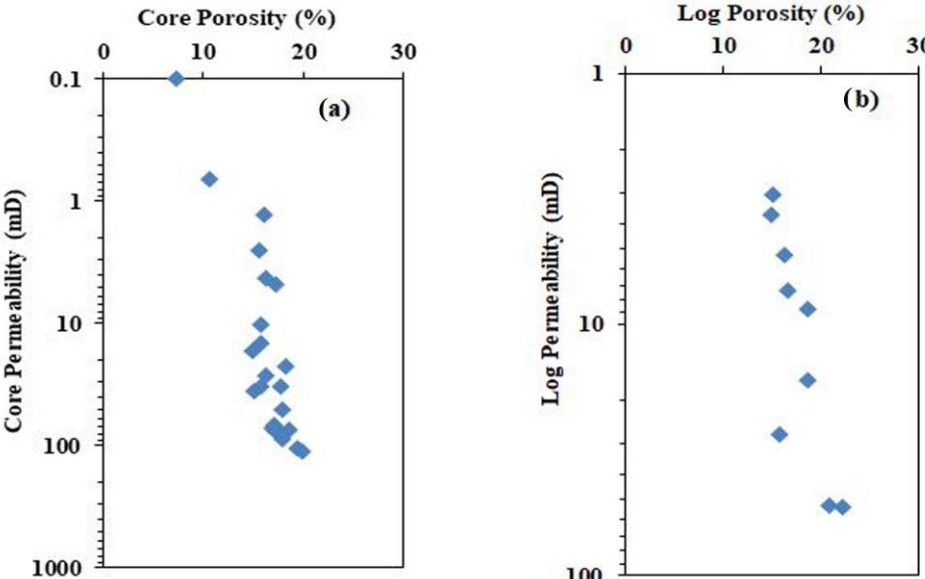

**Figure 7.** Relationship between (**a**) core permeability vs. core porosity (%); (**b**) log permeability (mD) vs. log porosity (%) of Srikail-3 well.

### 4.5. Diagenesis

#### 4.5.1. Compaction

Different diagenetic processes develop with the increase in the compaction rate, which has a substantial impact on the studied area. Diagenetic processes, such as packing rearrangement of the grains including long, concavo-convex and suture contacts (Figure 8a) and the thinning, bending and squeezing of the ductile grains, took place during shallow and intermediate burial (Figure 8a–c). Brittle grain fracturing is produced by deep burial mechanical compaction (Figure 8d). All of these diagenetic processes consequently change the bulk volume of rock. Depositional porosities are brought down to their existing levels in large part through compaction [19,20].

#### 4.5.2. Cementation

Clay Cement

The average amount of clay cement is 6.9%, of which 1.9% is authigenic chlorite. The predominant clay minerals are chlorite, illite-smectite/illite and minor kaolinite, as revealed by thin section, XRD and SEM. XRD patterns of the <2 μ clay fractions in these sandstones (Figure 9) indicate an overall relative chlorite content (25.2% to 29.4%), combined illite/illite-smectite content (37.5% to 49%), a kaolinite content (22 to 26.8%), and minor smectite content (2.8 to 10.4%).

Clay cement occurred in the form of pore-filling, grain coating and thin rim (Figure 10a,b). From SEM, it can be said that lath-like/crenulated shaped illite (Figure 10e) is one of the common clay minerals in this sandstone. Illite-smectite mixed layers are frequently found, and they often have crystals that resemble honeycombs and have a lath-like crenulated shape. Kaolinite appears as a stack of booklets (Figure 10e) in the secondary pores of feldspar grains. Authigenic chlorite appears as a thin rim, grain coating and pore-filling (Figure 10c,d). In most instances, it replaced biotite. Under SEM, chlorite comprises small irregular to pseudo-hexagonal platelets (Figure 10f) that are aligned perpendicular to the grain and occasionally wrapped by quartz overgrowths.

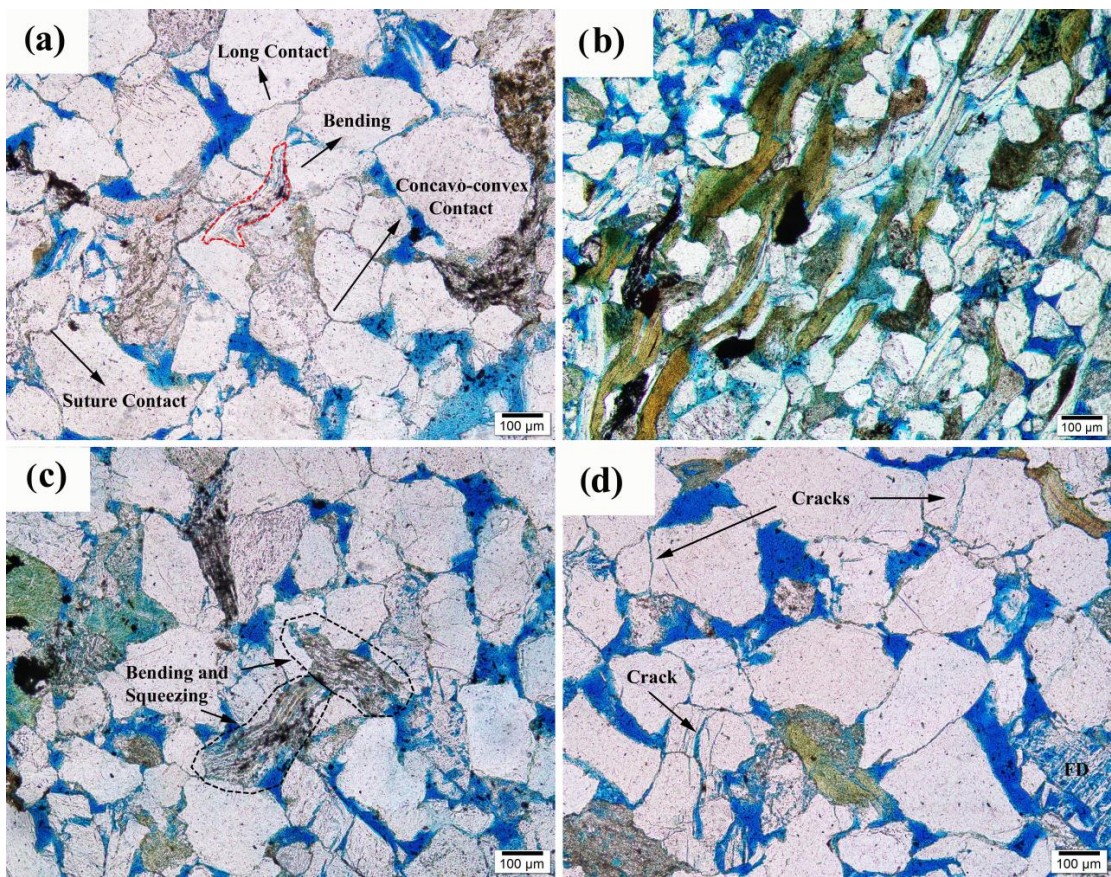

**Figure 8.** Photomicrograph showing the impact of the mechanical compression of the Srikail-3 well (**a**) long, concavo-convex, suture contact and bending of ductile grain at 3182 m depth; (**b**) ductile grain compaction (biotite) at 3181 m depth; (**c**) bending and squeezing of lithic grains at 3184.5 m depth; (**d**) a crack develops in brittle grains due to compaction, feldspar dissolution (FD) at 3183.5 m depth.

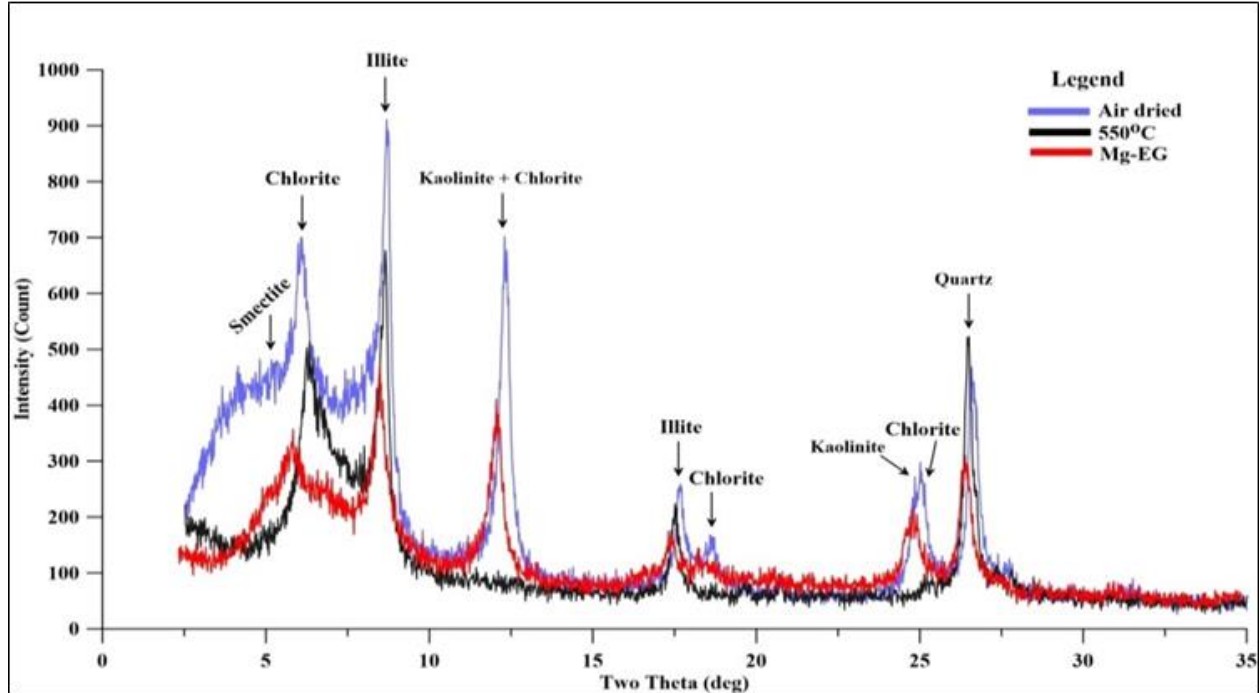

**Figure 9.** X-ray diffraction pattern of clay fraction of shale from Srikail-3 well sample no. (1c) at 3083 m.

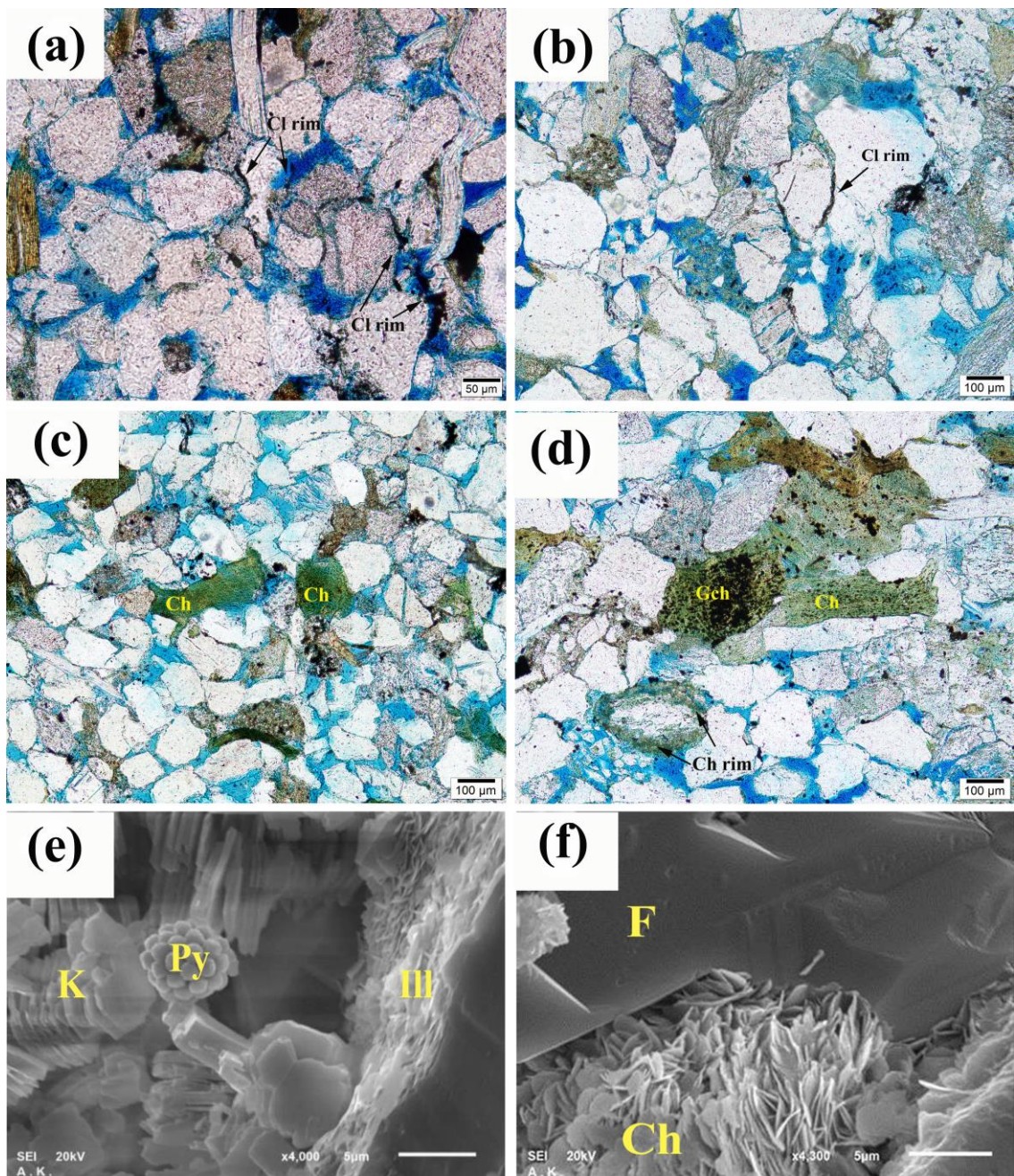

**Figure 10.** Photomicrograph showing (**a**) pore-filling and grain-coating clay at 3086.5 m depth; (**b**) clay rim at 3182 m depth; (**c**) pore- filling and grain coating chlorite at 3084.5 m depth; (**d**) chlorite rim and grain coating chlorite at 3184.5 m depth; SEM showing (**e**) kaolinite clay (K) with pyrite crystal (Py) and Illite (Ill); (**f**) pore- filling chlorite (Ch) and well-developed feldspar overgrowth (F) [4].

Carbonate Cement

Carbonate cement is one of the common diagenetic overprints (av. 2.5%) observed in the studied sandstones, and occurs as the poikilotopic (Figure 11a) and isolated pore-filling type (Figure 11b). However, the isolated pore-filling carbonate type (Figure 11b) is more common than the poikilotopic type. This poikilotopic calcite cement (~46%) forms an interlocking texture and locally replaces detrital quartz, feldspar, micas, and clay rich ductile grains at a depth of 3179.5 m.

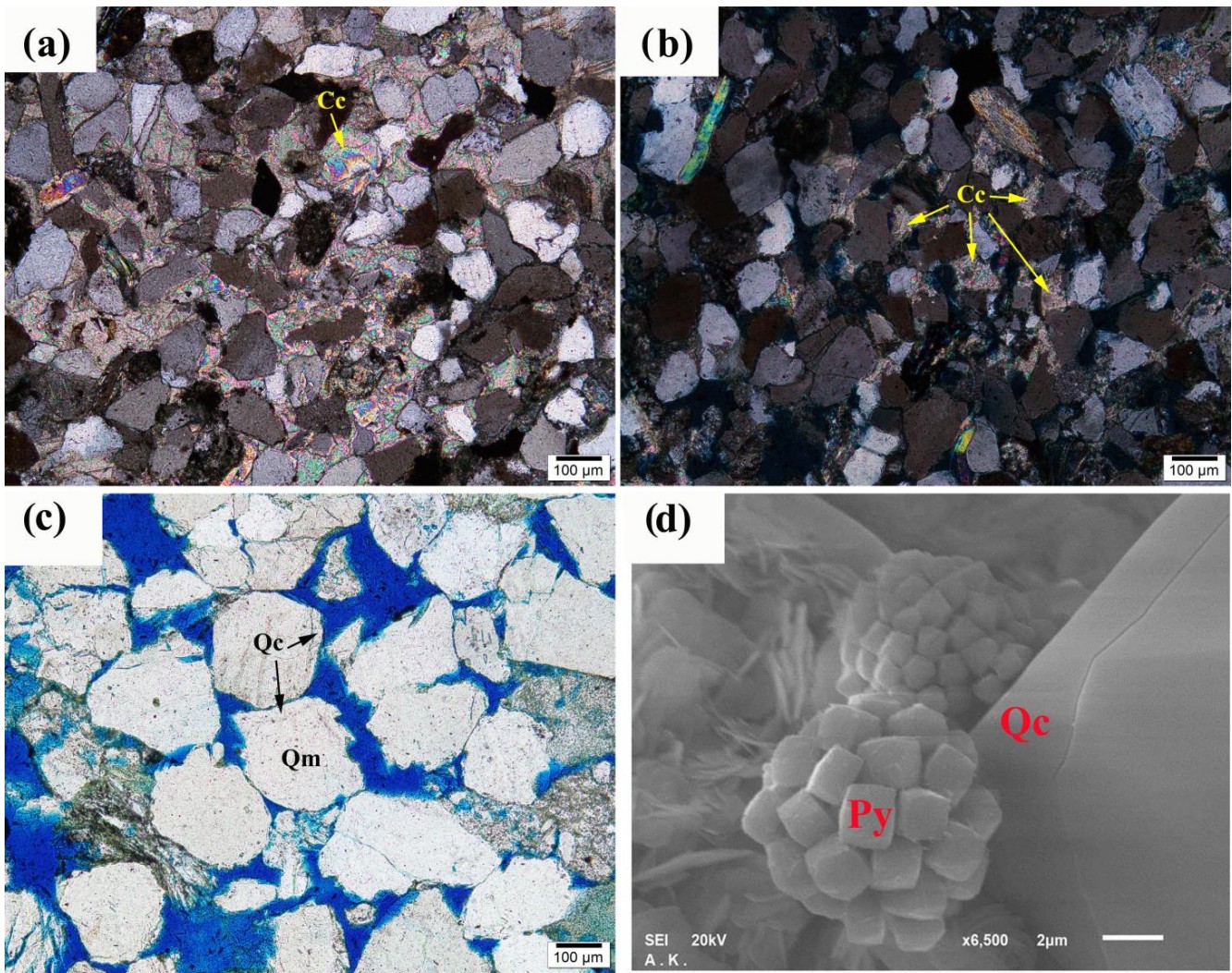

**Figure 11.** Photomicrograph showing (**a**) poikilotopic calcite cement (Cc) at 3179.5 m depth; (**b**) isolated pore-filling calcite cement (Cc) at 3085.5 m depth; (**c**) quartz overgrowth at 3182 m depth; (**d**) SEM shows framboidal aggregates of pyrite (Py) and quartz overgrowth (Qc) [4].

Quartz Cement

Quartz cementation is a signature of later diagenesis [21] that is subjected to pressure solution, and when super-saturation occurs, the quartz overgrowth forms as silica precipitation. It is a syntaxial quartz overgrowth that constitutes 0.19% of the samples studied. Although certain complete quartz overgrowths are also present, quartz cement often appears as small, isolated, euhedral outgrowths on detrital quartz grains (Figure 11c,d). In several instances, localized chlorite grains also hinder the formation of quartz overgrowth.

## 5. Discussion

### 5.1. Sequence of Diagenetic Events

The textural relationship of the thin section and SEM observations is used to establish the sequence of diagenetic events (Figure 12). The highest temperature limit for eodiagenesis is less than 70 °C, which is typically equivalent to approximately 2 km of burial. After eodiagenesis, mesodiagenesis takes place, with a temperature limit of >70 °C and a burial depth of >2 km [22]. The sediment was initially deposited in a basin and then gradually buried, and clay infiltration (Figure 10a) may have occurred. As sedimentation increases, compaction starts as a result of overburden pressure, which is an early diagenetic phenomenon that happens just after clay infiltration. Grain contacts are one

of the primary indicators of compaction effects. Long contacts, concavo-convex surfaces and suture contacts (Figure 8a) may indicate moderate to severe mechanical compaction of the sandstones [3]. Owing to the squeezing and extruding of ductile grains between rigid grains, this also has an impact on permeability [23]. As the diagenetic process moves forward, the basic processes of grain reorganization, a change in fabric or packing (such as simple cubic packing or hexagonal rhombohedral packing), ductile grain deformation (Figure 8b,c), fractures of brittle grain (Figure 8e) and pressure solution happen gradually. The textural interactions of the authigenic chlorite (Figure 10b,d) with other minerals show that they are early to late diagenetic components. The production of chlorite in the Neogene sandstones may have occurred with the replacement of detrital biotite or the transformation of antecedent infiltration or grain coating clay, or otherwise through a combination of these two processes [3]. Chlorites can also improve the compressive strength of rocks by filling the pores or coating the grains [24].

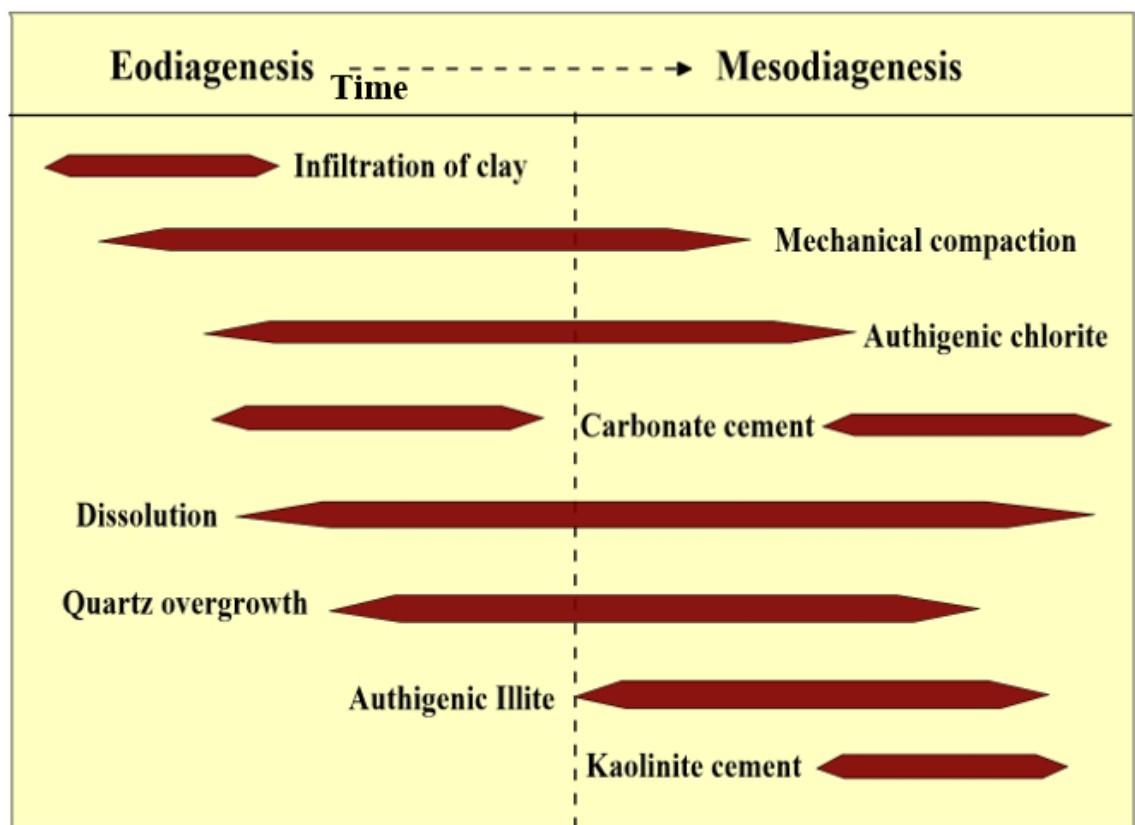

**Figure 12.** Paragenetic sequence of diagenetic features observed in the Neogene sandstones of the Srikail Gas Field.

Both poikilotopic and isolated carbonate cements (Figure 11a,b) are observed in the studied samples. The poikilotopic carbonate cement preserved an intergranular volume (IGV) of ~37% and filled relatively large pores between loosely packed framework grains as well as the partial replacement of grains, and is believed to have originated before considerable compaction took place.

The generation of the poikilotopic, pore-filling cements may have been considerably aided by the addition of $Fe^{2+}$ bicarbonate and hydroxyl ions to the pore fluids from nearby shale [25].

Poikilotopic carbonate cement from the Bengal Basin indicates that carbon was most likely derived from the thermal maturation of organic matter in adjacent shale during burial [3], and that meteoric pore waters were present during poikilotopic calcite growth [3].

In deeply buried sandstones, quartz overgrowths (Figure 11c,d) constitute a late diagenetic phase. The dissolution of feldspar by an acidic pore fluid, which may have

been combined with fluid from nearby strata and meteoric fresh water, is the cause of early quartz cement at low temperatures of between 60 °C and 80 °C. Organic acids may also dissolve feldspar during the early stages of hydrocarbon maturation at temperatures of between 80 and 120 °C because they lower the pH of the pore water [22,26]. Quartz cement can also be produced by the transformation of smectite to illite.

Most of the illite-smectite/illite minerals (Figure 10e) are known to be late-stage diagenetic minerals. Illite often originates at temperatures of more than 90 °C during progressive burial (mesodiagenesis) as a result of the alteration of depositional or infiltrating clays [22–27]. A source of potassium is needed for the illitization of smectite, which is most likely provided in this case via feldspar alteration [22–28]. In sandstones that are deeply buried, kaolinite also serves as an essential illite precursor.

It is thought that feldspar dissolution is connected to authentic kaolinite cementation, and it is an event of late mesogenesis which is indicated by thick, well-organized kaolinite booklets and blocky dickite (Figure 10e). Kaolinite is more likely to form when there is a poor K+/H+ ratio and few $SiO_2$ in the pore water [24] and references therein. Additionally, there is a late dissolution stage that produces secondary porosity by dissolving unstable mineral grains which is followed by the formation of a kaolinite cementing phase.

*5.2. Depositional Control on Reservoir Quality*

The quality of the reservoir in the Srikail Gas field is significantly influenced by depositional control. In the analyzed samples, sandstones of parallel-bedded, planar- and trough cross-bedded and massive sandstones in high energy distributary channels have the highest detrital quartz content, the lowest matrix content, and are fine- to medium-grained and well sorted. These high energy distributary channel deposits at depths from 3178 m to 3186 m in Srikail-3 offered larger pore and throat sizes and thus higher permeability compared to the finer grained sandstones of MFA (Ripple-laminated sandstone (Sr), wavy-bedding (Sw) and flaser-bedded (Sf) from depths of 3083 m to 3092 m (Figure 3 ). Core porosity and permeability have the lowest values at a depth of 3084 m, but maximum values are found at depths of 3181 m and 3182 m, respectively [4]. Higher shale content (Vsh) at depths of 3083 m to 3092 m lowered the quality of the D-Upper sandstone reservoir compared to the D-Lower sandstone and E-sandstone reservoirs. There are some shale layers (Figure 4f) within these sandstones that might work as baffle layers which have great effects on the permeability of a reservoir. With sedimentation during deposition in a low energy environment, the presence of depositional mud-flakes and clay drapes, matrix (Figure 13a) and ductile minerals (Figure 13b) such as muscovite and biotite in the reservoir sandstone decrease porosity and permeability by clogging interconnected pores, which has an impact on reservoir quality.

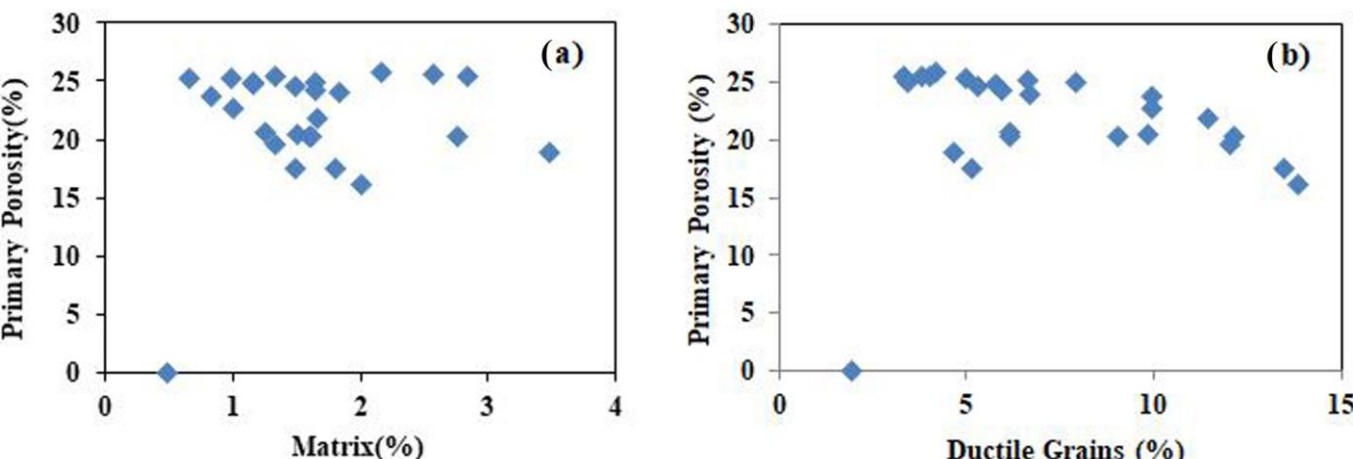

**Figure 13.** Relation between (**a**) primary porosity (%) vs. matrix (%) and (**b**) primary porosity (%) vs. ductile grains (%).

### 5.3. Diagenetic Controls on Reservoir Quality

5.3.1. Compaction Controls

Generally, with greater burial depths, the effect of compaction becomes more pronounced. Sandstones with finer grains typically experience more compaction than sandstones with coarser grains. The effect of compaction is high in the center of the fine grain sandstone, and the edges are where the effect has the least impact. Porosity and permeability had been reduced at the depth > 2000 m owing to grain rotation and rearrangement, grain bending, and ductile deformation of mechanically-weak grains (Figure 8a–c). Sometimes, fractures developed between the brittle grains due to compression, which may increase the permeability of the reservoir (Figure 8d). The plot of Houseknecht (1987) [29]) can be used to evaluate intergranular volume (IGV), cement concentration, and porosity to determine the relative impact of compaction and cementation (Figure 14). The result shown in Figure 14 indicates that there is higher compactional porosity-loss where there is a higher initial content of ductile grains. From this diagram, it can be concluded that mechanical compaction played a major role in reducing porosity of the sandstones.

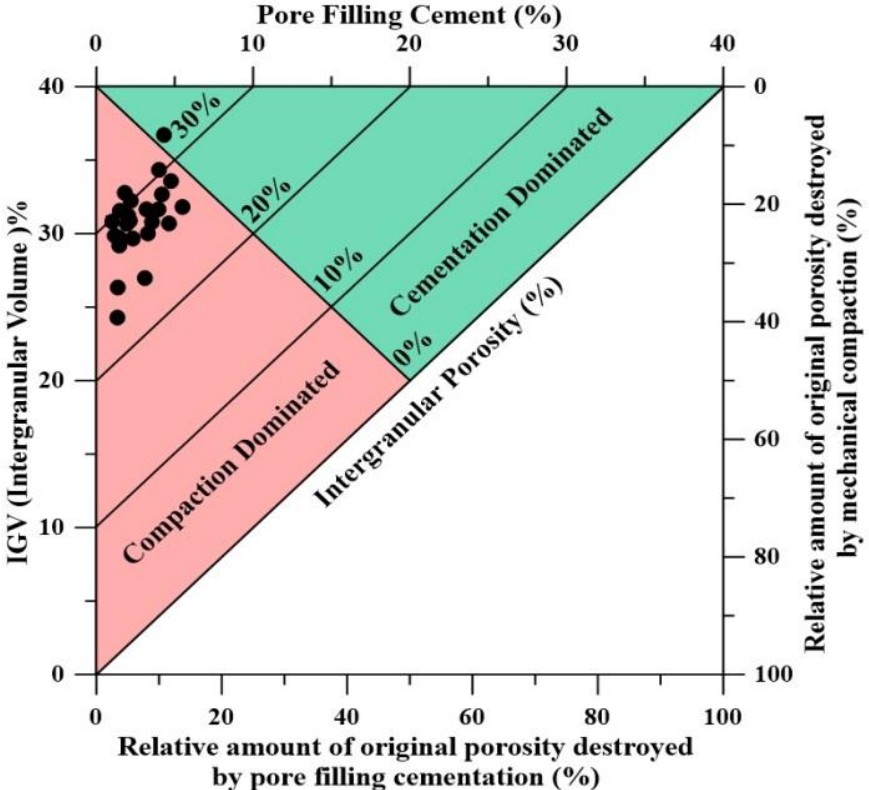

**Figure 14.** Plot of intergranular volume (IGV) versus cement of the Neogene sandstone in the Srikail-3 well after [29]).

5.3.2. Diagenetic Cement Controls

The second most dominant factor which plays a significant role in reservoir quality is clay cement. Pore-filling chlorite (Figure 10c) and illite-smectite/illite (Figure 10e) both reduce porosity and permeability, but their effects are more pronounced for permeability than for porosity [30]. The studied sandstones only contain trace amounts of quartz cement (Figure 11c,d), indicating that it has a limited impact on porosity and permeability reduction. The sandstones contain a lot of grain coating/rim chlorite cement (Figure 10d), which may prevent the precipitation of quartz cement and preserve porosity [3]. Calcite cement (Figure 11a,b), which frequently forms after burial and blocks pores or pore throats, has a strong negative connection with permeability and porosity. If the proportion of calcite is greater than 10%, the quality of the reservoir declines drastically due to the destruction of primary pores [31]. The reservoir quality is more significantly impacted by the poikilotopic

calcite cementation, which also exhibits a drastic decrease of porosity and permeability locally. The graphical representation of the relationship of different cements with core porosity and permeability is shown in Figure 15.

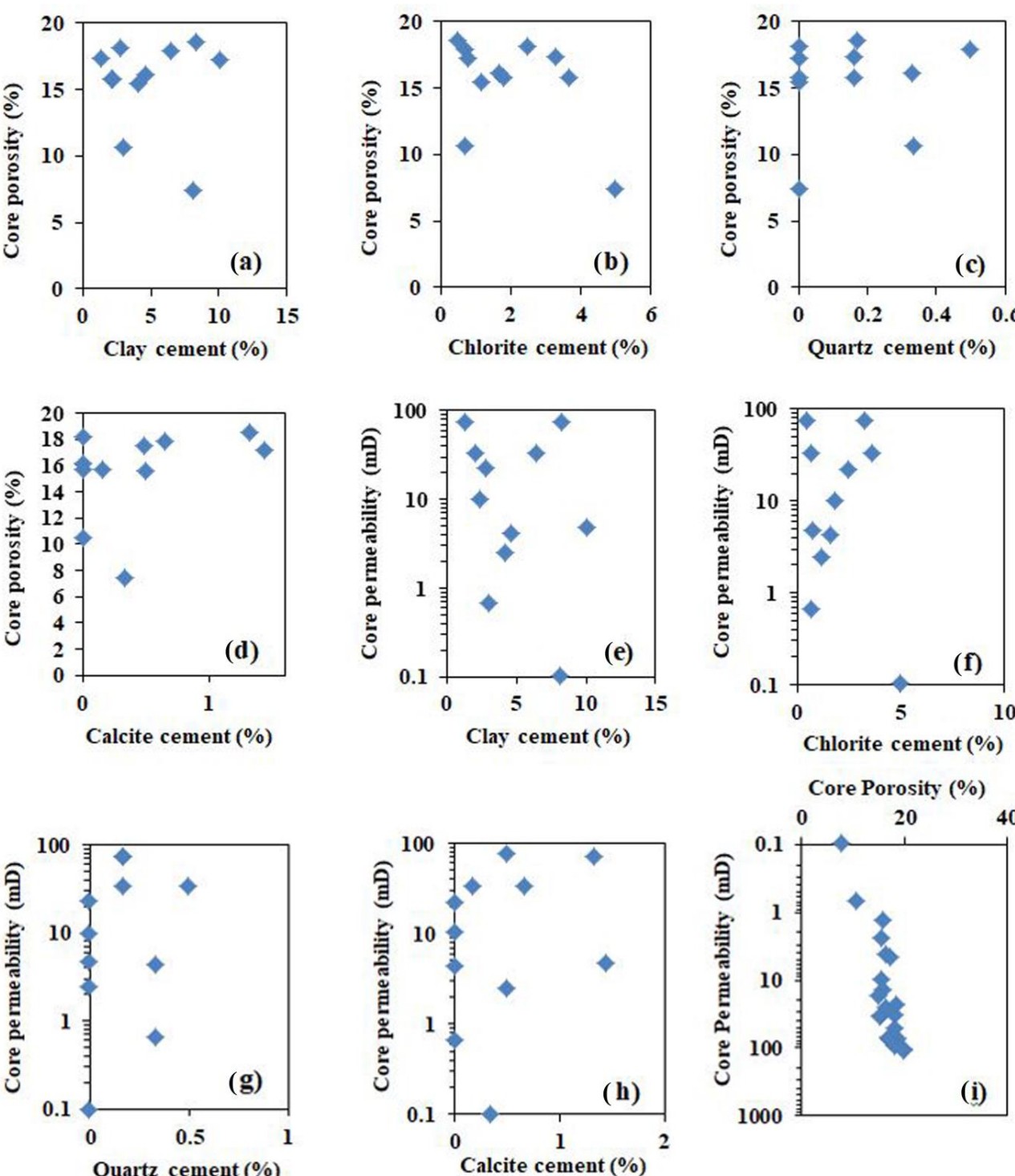

**Figure 15.** Relationship between (**a**) core porosity (%) vs. clay cement (%); (**b**) core porosity (%) vs. chlorite cement (%); (**c**) core porosity (%) vs. quartz cement; (**d**) core porosity (%) vs. calcite cement (%); (**e**) core permeability (mD) vs. clay cement (%); (**f**) core permeability (mD) vs. chlorite cement (%); (**g**) core permeability (mD) vs. quartz cement (%); (**h**) core permeability (mD) vs. calcite cement (%); (**i**) core permeability (mD) vs. core porosity (%).

### 5.3.3. Dissolution Controls on Reservoir Quality (Secondary Porosity)

Reservoir quality is significantly influenced by dissolution. As burial depth increases, the high mechanical compaction and cementation partially to entirely eliminates the primary porosity. In the process of diagenesis, acidic fluids are discharged from the source rock that react with the framework grains of the sandstones and can dissolve low-stability silicate minerals such as feldspar and unstable rock fragments (Figures 4e and 8d), which ultimately resulted in significant secondary porosity. In the studied sandstones, secondary porosities present as oversized pores (Figure 16a), moldic pores (Figure 16b) and microfractures within grains. The average secondary porosity is 6.8%, which suggests that dissolution has a considerable impact on reservoir quality in the studied sandstone from the Srikail-3 well.

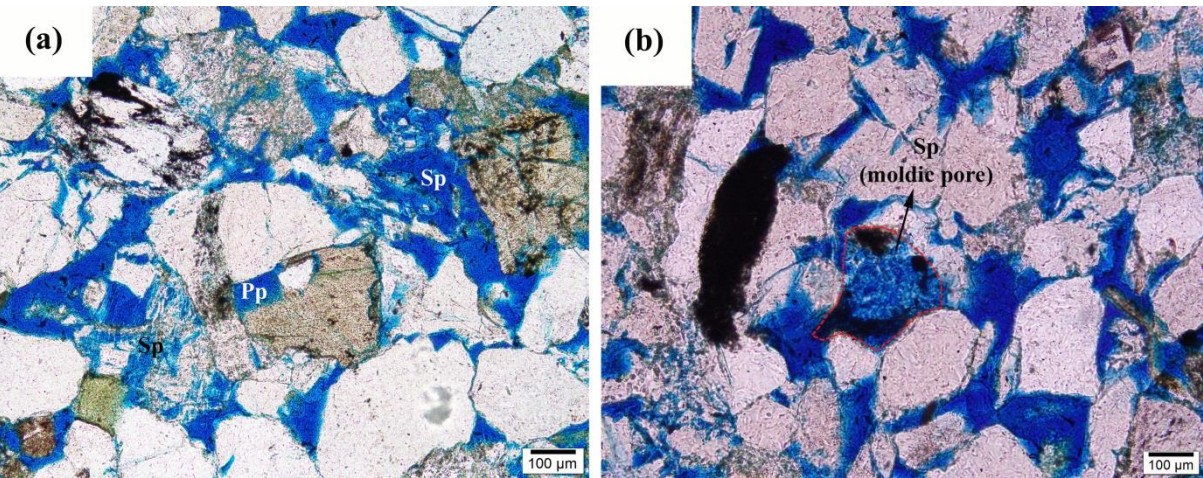

**Figure 16.** Photomicrographs showing (**a**) partial dissolution and oversized pore at 3182 m depth; (**b**) secondary porosity (Sp) (moldic) at 3181 m depth.

### 6. Conclusions

(i) The Neogene Surma Group of the Srikail Gas Field was deposited in a shallow marine tide-dominated delta system.

(ii) Reservoir sandstones in the Srikail Gas Field are subarkose to sublitharenite in nature and have good porosity (thin section primary porosity-21%, log porosity 18%, core porosity 16%) but poor to moderate values of permeability (0.1 mD–76 mD with an average of 31 mD).

(iii) Better quality reservoirs are characterized by moderate to well sorted medium to coarse grained planar and trough cross-bedded sandstones having porosity (av. 16.60%) and permeability (av. 45.22 mD) at depths of 3178 to 3190 m.

(iv) The porosity reduction in the Neogene Sandstone was primarily caused by mechanical and ductile grain compaction. Chlorite grain coating prevents quartz cementation and preserves porosity. Poikliotopic carbonate cement drastically destroyed the porosity and permeability of the reservoir sandstone locally. The dissolution of unstable mineral grains has a positive impact on the porosity and permeability of a reservoir.

(v) Overall, it can be concluded that the Neogene Surma Group in the Srikail Gas Field offers a moderate to good quality reservoir. The implications of this work can increase understanding of reservoir potential and help in the exploration of hydrocarbons in an efficient way. Future studies of 3D high-resolution seismic and detailed sequence stratigraphy are recommended for better exploration of the subsurface structure and hydrocarbon-bearing zones.

**Author Contributions:** M.A.: Sample collection, data analyses, manuscript writing; M.J.J.R.: Laboratory work, manuscript writing; M.M.: Data analyses; D.H.: Data analyses; F.K.: Data analyses. All authors have read and agreed to the published version of the manuscript.

**Funding:** This study was sponsored by National Science and Technology Fellowship 2021–22 and Jahangirnagar University Project 2021–2022.

**Data Availability Statement:** Not applicable.

**Acknowledgments:** We would like to express our gratitude to the Chairman of Petrobangla and the Managing Director of BAPEX for their kind approval for core analysis and the Laboratory of Soil Geology and Environment, Institute of Geology and Geophysics, Chinese Academy of Sciences (CAS) for their laboratory support. The first author would also like to acknowledge the National Science and Technology Fellowship of the Government of the People's Republic of Bangladesh and Jahangirnagar University for providing the research funding.

**Conflicts of Interest:** The authors declare that they have no conflict of interest.

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
