# Peer review of "Depositional and Diagenetic Controls on Reservoir Quality of Neogene Surma Group from Srikail Gas Field, Bengal Basin, Bangladesh"

_minerals, doi:10.3390/min13020283_

Round 1
Reviewer 1 Report
The article is well written and suggested with minor revision; Detailed comments are listed below.
The abstract provides a good overview of the study's objectives and methods, but it could be more specific about the results and conclusions. Suggestion: Include specific data and key findings in the abstract to give readers a better sense of the study's significance.
The introduction clearly lays out the background and motivation for the study, but it could benefit from more information on the specific geology and tectonics of the Bengal Basin and the Srikail Gas Field. The study focuses on static properties; dynamic properties should be introduced as well the articles below are suggested to be consulted as starting points. Status and Outlook of Oil Field Chemistry-Assisted Analysis during the Energy Transition Period. Energy & Fuels 2022. Permeability measurement of the fracture-matrix system with 3D embedded discrete fracture model. 2022. Suggestion: Add a brief section on the regional geology and tectonics to provide context for the study.
The methods section is well-organized and provides detailed information on the data collection and analysis techniques used, but it could be more concise. Suggestion: Consider simplifying the language and removing any unnecessary details to make the methods more readable.
The results section presents a wealth of data, but it could be more effectively organized and presented. Suggestion: Use tables, figures and diagrams to help readers understand the key results and their significance.
The discussion and conclusion sections could be more explicit about the implications of the study's findings for petroleum production and exploration in the Bengal Basin. Suggestion: Provide specific recommendations for future research and exploration activities based on the study's results.
The paper overall is well-written and well-researched, but it could benefit from more interpretation of the data and more discussion of the implications of the study's findings. Suggestion: Provide more detail on how the data and findings relate to the study's objectives and questions, and what the implications are for the field of study.
Author Response
Response to Reviewer 1 uploaded

Reviewer 2 Report
Reservoir quality is important for oil and gas production. Logs and core analysis data were used to evaluate the reservoir quality of the study area in manuscript, and the control factors on sequence of diagenetic events and depositional were also discussed.
Some comments as following are provided for author to revise manuscript.
1. Line 132, 4.2 Lithofacies and Facies Association. What is the criterion to identify the Facies. Generally, lithofacies is characterized with logs response without core.
2. Line 215, 4.4 Petrophysical Properties. How to calculate porosity, permeability and shale contents with logs, please add the petrophysical parameters model.
3. In Table 4, the big difference is total porosity of Thin Section and core porosity, why? Total porosity of Thin Section is reliable in Figure 13?
4. “Crack” is better than “Facture” in Figure 8d).
5. Line 451, “v) Overall, it can be concluded that the Neogene Surma Croup in the Srikail Gas Field offers moderate to good quality reservoir.” Why? This issue is not discussed in the text.
6. Figure 1, Figure 2 and Figure 6 are not clear, especially text in the figure.
Author Response
Response to Reviewer 2 uploaded

Reviewer 3 Report
With a variety of analytical methods (wire-line logs, core analysis, petrography, XRD and SEM), this MS tries to reveal the depositional and diagenetic controls on reservoir quality of the Neogene Surma Group at the Srikail Gas Field, Bengal Basin, Bangladesh. It is a meaningful work, which might have a certain guiding significance for oilfield production. There are also some obvious problems that need to be corrected before being published.
1) Please explain the innovation of this article.
2) The conclusion is too long and needs to be refined.
3) Lithofacies abbreviations are ambiguous (Line 133-138, 146-154, and 166-172). The same abbreviation refers to different full names. It is recommended to use the same full name or different abbreviations to make it consistent.
4) Referring to the work of predecessors (Sun et al., 2022), it is suggested to supplement the work on inclusions to make the diagenetic sequence more convincing.
Sun, D., Liu, X., Li, W., Lu, S., He, T., Zhu, P., Zhao, H., 2022. Quantitative evaluation the physical properties evolution of sandstone reservoirs constrained by burial and thermal evolution reconstruction: A case study from the Lower Cretaceous Baxigai Formation of the western Yingmaili Area in the Tabei Uplift, Tarim Basin, NW China. Journal of Petroleum Science and Engineering 208, 109460. https://doi.org/10.1016/j.petrol.2021.109460
5) The content layout needs to be improved to make sentences continuous (Line 435-438). Remove extra spaces (Line 16).
6) Figure 1 is very vague. It is recommended to draw a vector diagram.
Author Response
Response to Reviewer 3 uploaded

Reviewer 4 Report
This manuscript carries out a study on Depositional and Diagenetic controls on Reservoir Quality of Neogene Surma Group from Srikail Gas Field, Bengal Basin, Bangladesh.
The following comments and suggestions should guide the authors to seriously revise the paper.
1. The manuscript needs the following changes to reach the standard of publication: (1) improve figures, (2) improve analysis and discussion, (3) improve the expression of innovation and research significance.
2. The language is fine and only need to check for minor grammar or spell errors
3. According to the title of the paper (Depositional and Diagenetic controls on Reservoir Quality of Neogene Surma Group from Srikail Gas Field, Bengal Basin, Bangladesh), the control of sedimentation and diagenesis on reservoir quality should be the focus of this paper. However, in the text, the author has described diagenesis and its impact on reservoir quality for a long time, while the impact of sedimentation on reservoir quality has been mentioned less. It is suggested to supplement the discussion on the influence of sedimentation on reservoir quality.
4. In the paper, only porosity, permeability, water saturation and other physical parameters of the reservoir are used to represent the reservoir quality, which cannot fully represent the reservoir quality. It is suggested to add other parameters that can represent the reservoir quality, or change the research content of the paper to the control of sedimentation and diagenesis on the reservoir physical properties.
5. In this paper, the porosity and permeability measured and calculated by test and logging methods are used respectively. Do the authors want to compare the differences or common characteristics of the parameters obtained by these two methods? Why and what is the purpose of using these two methods?
6. The paper lacks qualitative and quantitative analysis of reservoir pore types.
7. Water saturation can indicate that gas-bearing sandstone has good hydrocarbon saturation——its source should be cited.
8. The resolution of Figure 1 needs to be improved.
Figure 2 is deformed, and the scale should be changed to redraw.
Figure 3: The scale is not clear, and the picture should be perpendicular to the observation plane.
Figure 4/10/15/16. The picture layout needs to be improved.
Figure 4c-d: There is no scale in the figure.
9. References: lack of latest research progress.
10. The paper should further strengthen the explanation of innovation points and new understanding in order to better attract readers.

Author Response
Response to Reviewer 4 uploaded

Reviewer 5 Report
It is an interesting paper that provides insight into the petrologic properties of a particular reservoir. It has a special value for researchers studying other reservoirs, as the paper provides a guideline for studying sandstones in a reservoir. The relationships they establish between processes and parameters are very interesting and useful.
However, several aspects should be clarified for it to be useful and for the interpretations to be well understood.
In addition, I think the paper should be more self-critical and the authors should better evaluate their results. For example, the study is based on a few meters of cores, but the interpretation is made for the whole formation under study. They do not explain or justify on what basis they make this extrapolation in the interpretation. Perhaps what they should do is to interpret but indicate the limitation of their observations. Even if their interpretation is limited in space, they are still good, but they should clarify it and, above all, explain well or rectify some aspects that I indicate below:
- Lines 89-93. In this paragraph nothing is said about the “Upper Marine Shale” formation which is of the Surna group and is assigned a Miocene age (Ref Table 1).
- In Table 1 Bokabil Fm is Late Miocene, and is Upper Marine Shale Miocene? It will be Late Miocene or Pliocene, if Bokavil Fm is Late Miocene, or is Bokavil early Miocene? How is this?
- Lines 126-131. These lithological zones should be indicated in Figure 2.
- In Figure 3, b) and c) are identical, it is the same core. I guess it will be too late to take a more detailed picture of the structures, so that the laminations can be seen. These photos are good to see the type of material that has been investigated, but they are not good to see structures.
- Line 174. It is interesting to state the number of the thin sections that has been studied
- Lines 161-164. It should be noted that the interpretation is made on the basis of the samples studied, which are limited, as 8 meters are studied in detail and the formation is 300 meters long. Can it be justified that the interpretation can be extrapolated to 300 m? If so, and because
- Line 180-195. The 32-51% quartz, 3,1% feldspar, 9% lithics does not sum to 100% to use the triangular plot. And the values in this triangular plot (Fig 5) is not the same. To be subarkose or sublitharenite it would have to have much more quartz. And shouldn't the position of the sandstone component values for the two triangles be the same? These data should be clarified.
- Line 198. I don't understand the meaning of this sentence. What does it mean?
- Figure 4. The formatting of figure 4 needs to be improved to make it more visible, the lettering needs to be better and the scale should not cover anything that needs to be pointed out with an arrow.
- Figure 6. The meaning of each acronym for the different tests, e.g. shale volume (VSH-GR) or gamma ray, should be indicated at the bottom of Figure 6.
- Figure 12. In figure 12, could the dissolution process be indicated?
- Line 365. This layer of clay, as seen under a microscope, is very thin (0.4 mm). For its effectiveness in permeability, it is necessary to talk about thicker layers or very thin but frequent layers, which can result in a wacke sandstone (with matrix). Please, explain this better, so that the value of the presence of clay, at millimeter levels (or about thicker levels), can be understood.
- Lines 421-424. The authors generally speak of dissolution of grains by the acidity of fluids at depth, in the example at 3183 m (Fig. 4). However, at 3179 m they speak of carbonate cementation (Fig. 11). Is this due to the passage of fluids through specific zones, at only 4 m? If we are talking about acid fluids, they cannot be saturated in calcium carbonate. Are these two processes occurring at different times? Can the dissolution be placed within the scheme of figure 12?
- Conclusions:
The conclusion that these are marine deposits is not supported by new data, or are there new data provided in this work to support this conclusion?
In the end, an assessment of the quality of the storage rock is made, although it should be pointed out that the study is limited because it is only done on the basis of very few samples within a very powerful formation, isn't it? The study is valid and gives good results, it is applicable to other storage rock studies, although its extrapolation to the whole formation is a bit risky.
Author Response
Response to Reviewer 5 uploaded

Round 2
Reviewer 3 Report
The authors have modified the MS according to the suggestions. And it is acceptable
Author Response
Reviewer 3

Reviewer 4 Report
It is suggested to add qualitative and quantitative analysis of reservoir pore types.
Author Response
Reviewer 4

Reviewer 5 Report
I have reviewed the new manuscript and the responses given by the authors. The modifications made are satisfactory and some omissions have been clarified and completed.
Author Response
Reviewer 5
